# Low grade intravascular hemolysis associates with peripheral nerve injury in type 2 diabetes

**Sylvain Le Jeune**[1,2], **Sihem Sadoudi**[1], **Dominique Charue**[1], **Salwa Abid**[1], **Jean-Michel Guigner**[3], **Dominique Helley**[1,4], **Hélène Bihan**[5], **Camille Baudry**[6], **Hélène Lelong**[7], **Tristan Mirault**[1,8], **Eric Vicaut**[1,9], **Robin Dhote**[2], **Jean-Jacques Mourad**[10], **Chantal M. Boulanger**[1], **Olivier P. Blanc-Brude**[1] *

**1** Université Paris Cité, INSERM, Paris Center for Cardiovascular Research-ParCC, Paris, France, **2** Service de Médecine Interne, AP-HP, Hôpital Avicenne, Bobigny, France, **3** Institut de Minéralogie, de Physique des Matériaux et de Cosmochimie, Sorbonne Université, Paris, France, **4** Service D'hématologie Biologique, Hôpital Européen Georges Pompidou, AH-HP, Paris, France, **5** Service de Diabétologie, Endocrinologie et Maladies Métaboliques, AP-HP, Hôpital Avicenne, Bobigny, France, **6** Service de Diabétologie, Endocrinologie et Nutrition, Hôpital Paris Saint-Joseph, Paris, France, **7** Unité HTA, Prévention et Thérapeutiques Cardiovasculaires, Hôtel Dieu, AP-HP, Paris, France, **8** Service de Médecine Vasculaire, Hôpital Européen Georges Pompidou, AH-HP, Paris, France, **9** U.R.C. Lariboisière-Saint Louis, AP-HP, Paris, France, **10** Service de Médecine Interne, Hôpital Paris Saint-Joseph, Paris, France

* olivier.blanc-brude@inserm.fr

**Data Availability Statement:** All relevant data are within the manuscript and its Supporting information files.

**Funding:** Project 'RépaDia' of the Fondation de France to OBB, Project 'BiFace' of the Fondation

## Abstract

Type 2 diabetes (T2D) induces hyperglycemia, alters hemoglobin (Hb), red blood cell (RBC) deformability and impairs hemorheology. The question remains whether RBC breakdown and intravascular hemolysis (IVH) occur in T2D patients. We characterized RBC-degradation products and vesiculation in a case-control study of 109 T2D patients and 65 control subjects. We quantified heme-related absorbance by spectrophotometry and circulating extracellular vesicles (EV) by flow cytometry and electron microscopy. Heme-related absorbance was increased in T2D vs. control plasma (+57%) and further elevated in obese T2D plasma (+27%). However, large CD235a+ EV were not increased in T2D plasma. EV from T2D plasma, or shed by isolated T2D RBC, were notably smaller in diameter (-27%) and carried heme-related absorbance. In T2D plasma, higher heme-related absorbance (+30%) was associated to peripheral sensory neuropathy, and no other vascular complication. In vitro, T2D RBC-derived EV triggered endothelial stress and thrombin activation in a phosphatidylserine- and heme-dependent fashion. We concluded that T2D was associated with low-grade IVH. Plasma absorbance may constitute a novel biomarker of peripheral neuropathy in T2D, while flow cytometry focusing on large EV may be maladapted to characterize RBC EV in T2D. Moreover, therapeutics limiting IVH or neutralizing RBC breakdown products might bolster vasculoprotection in T2D.

## Introduction

The number of adults with type 2 diabetes (T2D) has more than doubled worldwide in the past 30 years and continues to increase exponentially. Five million deaths were attributed to T2D in 2017 [1]. The health burden is largely due to cardiovascular diseases and microvascular

pour la Recherche Médicale to OBB, Project 'Cardannex' of Inserm Transfert to OBB. The funders had no role in study design, data collection and analysis, decision to publish, or preparation of the manuscript.

**Competing interests:** The authors have declared that no competing interests exist.

complications associated to T2D, fostered by poor glycemic control. However, hyperglycemia alone, independently of traditional cardiovascular risk factors, does not explain more than 25% of the excess risk in T2D [2], suggesting the involvement of other pathophysiological processes.

Questions were raised whether red blood cell (RBC) exert a pathological role in T2D. An array of studies reported glyco-oxidative alterations of RBC membranes, increased RBC osmotic fragility, reduced lifespan and degraded hemorheological properties in T2D, as reviewed by Turpin, Pernow and colleagues [3, 4]. RBC remodeled during T2D can compromise endothelial vasodilation *in vitro* via increased arginase activity, ROS and peroxynitrite production [5–7]. They can also modify the morphology of cultured endothelium [8]. The rigidity of T2D RBC was associated to nephropathy [9] and observed during retinopathy [10]. However, the independent impact of T2D RBC on the vasculature remains difficult to assess in patients, and the question remains, whether T2D promotes RBC destruction within the circulation, a phenomenon termed intravascular hemolysis (IVH).

In clinical practice, IVH is identified by indirect biomarkers including anemia, reticulocytosis, high lactate dehydrogenase (LDH) and bilirubin levels, and low haptoglobin levels, sometimes combined with hemoglobinuria. These indirect markers are modulated by inflammation independently of IVH and lack sensitivity. Most, like hemoglobinuria, are ignored until they become conspicuous. Indirect markers of IVH are rarely monitored in T2D patients.

In cell biology, IVH is considered as a leak of RBC contents into plasma (Heme-containing hemoglobin (Hb), globin-free heme, oxidized forms of heme) and membrane fragmentation, with extracellular vesicles (EV) shedding. These moieties can combine into heme-loaded EV during IVH [3, 11, 12]. Several flow cytometry (FACS) studies reported elevated plasma levels of circulating large EV in T2D and metabolic syndrome patients, primarily shed by platelets, monocyte and endothelial cells [13]. Large RBC-derived EV were also increased up to 4-fold in studies comparing obese T2D patients to obese or lean, non-T2D controls [14–16], or lean T2D patients to healthy subjects [17–19]. However, they were unchanged in another study of T2D patients vs. controls [20], or in T2D vs. non-T2D patients with moderate/high cardiovascular risk [21]. Besides, T2D is associated to increased heme-oxygenase-1 activity and reduced bilirubin plasma levels [22, 23]. These data are difficult to reconciliate. Consequently, RBC breakdown products are hardly ever considered in clinical practice, whereas many data support a possible link between IVH and vasculopathy.

In plasma, cell-free Hb and heme scavenge NO [24], stimulate pro-oxidant, cytotoxic and inflammatory reactions in the vascular endothelium and compromise vasodilatation [25, 26]. Intravenous administration of Hb provokes vascular damage [27] and cell-free Hb can be linked to vascular complications during IVH [24, 27]. Plasma Hb and heme constitute relevant hallmarks of IVH. Their measurement in biofluids can be implemented using the unique spectrophotometric characteristics of heme, with absorbance peaks at 398 and 575 nm (Soret bands) [12, 28–32]. The levels of plasma Hb and heme have never been accurately reported in T2D. We aimed to 1/ establish whether IVH occurs in T2D, by comparing Soret band-related absorbance and FACS quantification of EV in the plasma of T2D vs. control patients; and 2/ investigate these biomarkers as potential indicators of vascular injury.

## Materials and methods

### T2D cohort 'Diabelyse'

'Diabelyse' was a biological collection registered with the national French ministry of health (n˚DC-2011-1480). We included 109 adult T2D patients in the diabetology unit of Avicenne

Hospital. Demographic, clinical, laboratory and treatment data were collected, including cardiovascular risk factors, micro- and macrovascular complications (S1 Table). 65 non-diabetic control volunteers were enrolled in Centre d'Examens de Santé (Bobigny) and Vascular Medicine Units of Avicenne, Hôtel-Dieu and European Georges Pompidou Hospitals. Eligibility criteria (Supporting information) were recorded and samples anonymized. We performed a case-control study nested in the biological collection. We obtained ethical approval from committee CPP-IDF-VII on 01/02/2012. The non-diabetic group included subjects with cardiovascular risk factors, mainly hypertension (HTN) (56%) and dyslipidemia (34%), in order to better illustrate the relative effect of T2D. We identified subgroups of patients with obesity, HTN and dyslipidemia. Informed written consents were obtained from all participants. The study design, within the biological collection, did not include patient matching. Blood was collected on citrated tubes according to MISEV-2018 guidelines [33]. Platelet-free plasma was prepared by double centrifugation within 4h and stored at -80°C (Supporting information).

## Circulating biomarkers of IVH

Cell-free heme was quantified by spectrophotometry, using its spectral characteristics and Soret band absorbance, main and secondary peaks, as reviewed by Alayash et al. [34, 35]. We selected a wavelength corresponding to heme (398 nm; total) and protein-coordinated heme (575 nm), ie. heme coordinated to its major partner in RBC to form Hb [28, 36]. We subtracted background values at 650 nm, a wavelength at which Hb and its degradation products do not absorb light. This technique detects protein-free and protein-bound heme, and limits any contribution of turbidity [12, 30–32]. We did not attempt to specify the different forms of ferrous, ferryl and deoxy-Hb which are all expected to contribute to Abs575 [34]. The spectral characteristics of Hb may be shared with other hemoproteins, but few other than Hb are abundant enough in the circulation to contribute to Soret band absorbance in large proportions. 67 T2D vs. 36 control plasma were depleted of EV by ultracentrifugation (20500-g, 4h) to estimate EV-associated Hb and heme [12].

## Flow cytometry, nanoparticle tracking and cryogenic transmission electron microscopy

We adhered to MISEV-2018 guidelines and used a panel of specific techniques to characterize EV [33]. We measured circulating RBC membrane fragments or large EV (microvesicles), by FACS after double-labeling with anti-CD235a (RBC-specific glycophorin-A; Becton Dickinson) and phosphatidylserine (PS)-binding annexin-V (Roche Diagnostics), as we published previously [12, 31, 32]. FACS was performed in runs of 30 samples including patient groups and controls. We analyzed plasma EV and RBC-EVs released *in vitro* using cryogenic transmission electron microscopy (CryoTEM; n = 6–8). Carbon membrane grids (Quantifoil Micro Tools, Germany) were loaded with 10 μl of EV, blotted and snap-frozen in liquid ethane. Plasma EV were concentrated 10-fold by UC (25000-g, 4h), while RBC supernatants were analyzed neat. Vitreous ice films were placed in a cooled Gatan 626 cryoholder, inserted into a LaB6 JEOL JEM2100 (JEOL, Japan) cryomicroscope operating at 200 kV, with a JEOL Minimum Dose System. Images were recorded at -180°C with an ultrascan 1000 CCD camera (Gatan, USA). EV were identified as circular objects with 40–3000 nm diameters and electron-dense phospholipid bilayers. Objects without these characteristics, like lipoproteins and chylomicrons were ignored. 200 to 550 EV were measured individually across 30–100 frames, using Image-J software (National Institutes of Health, USA). Size distributions of RBC-derived EV were confirmed by nanoparticle tracking analysis (NTA) using a Nanosight LM14 (Malvern, France).

## Cell culture and vesiculation *in vitro*

RBC were sorted on Granulosep$^®$ (Eurobio) density gradients (700-g, 30 min at RT) (leukocytes, granulocytes, platelets removed) and adjusted with PBS to 40% hematocrit (equivalent). Vesiculation was stimulated by adding $Ca^{2+}$ (5 mM) and ionophore A23187 (2 μM) [12]. Alternatively, RBC (1 mL) were placed into a Dounce homogenizer and 50 strokes were applied to create shear stress. RBC were pelleted by bench-top centrifugation and supernatants collected. CryoTEM and NTA analyses of EV shed by RBC *in vitro* were performed in intact supernatants, concentration proved unnecessary.

For functional studies, large RBC EV were pelleted by UC and quantified by FACS and NTA as above. Human umbilical cord endothelial cell (HUVEC) monolayers were cultured to confluency and treated with RBC EV. Some EV were preincubated for 1h with recombinant hemopexin (1 μM) or annexin-A5 (10 μg/mL). HUVEC were pre-treated with DPI (10 μM), Ro-31-8220 (1 μM), or Gö-6976 (1 μM). RBC EV (25–1000 x10e5 EV/mL) were applied onto HUVEC cultures for 2h. Radical oxygen species production was evaluated with 5-(and 6-)carboxy-29,79-difluoro-dihydro-fluorescein diacetate ($H_2$DFF-DA) for 30 minutes [12].

To reveal differences in EV surface properties, we used a Calibrated Automated Thrombogram$^®$ platform (Diagnostica Stago, USA) and measured the thrombin-activating potential of RBC-derived EV. This enzymatic activity is dependent on surface charge, phosphatidylserine and Ca2+-dependent recruitment of serine proteases. Exogenous artifices were eliminated from routine hospital protocols in order to reveal EV-supported reactions. Only RBC-derived EV (10e9 EV/mL) were added to control plasma depleted of endogenous EV (25000-g, 4h). Synthetic phospholipids or recombinant tissue factor were excluded.

## Laboratory parameters in 'Diabelyse'

Patient laboratory data were gathered in standard follow-up procedures by the biology platforms of AP-HP Hospitals and CPAM [37], including RBC-related parameters (hemoglobinemia, hematocrit, mean corpuscular volume (MCV), mean corpuscular Hb concentration (MCH) and indirect markers of IVH (aspartate aminotransferase (AST), alanine aminotransferase (ALT), LDH, Bilirubin and iron. Other blood data helped characterize T2D status, including fasting glycemia, HbA1c and C-peptide levels. Dyslipidemia was defined by the presence of one of four criteria: Triglyceride levels >1.5g/L, LDL-cholesterol >1g/L (T2D patients) or >1.3g/L (controls), HDL-cholesterol <0.4g/L, lipid-lowering therapy. We used the LDL-cholesterol threshold of 1g/L, recommended in 2011 ESC-EAS guidelines [38]. LDL-cholesterol levels were >0.55g/l in 94% of our T2D patients. Therefore, the 2011 ESC-EAS guidelines were more discriminative than the LDL-cholesterol threshold target proposed for T2D patients in 2019 ESC-EAS guidelines [39].

## Micro- and macrovascular injury in 'Diabelyse'

The presence of micro- and macrovascular injury [40] was screened across T2D patients. Nephropathy was evaluated by measuring albuminuria, proteinuria and creatininuria in random urine samples, and estimating glomerular filtration rates (eGFR) with the Chronic Kidney Disease Epidemiology Collaboration (CKD-EPI) equation after determination of plasma creatinine levels. Nephropathy was defined as urinary albumin-creatinin ratios ≥20 mg/g and/or stage 3 chronic kidney disease (CKD) (eGFR <60 mL/min/1.73 m$^2$). Retinopathy was detected by trained ophthalmologists via fundoscopy, and graded using the classification tables of Société Francophone du Diabète (SFD). Peripheral sensory neuropathy was assessed clinically by the presence of suggestive symptoms (DN4 questionnaire), a vibration perception threshold of 25V or more, and/or the inability to feel a 10-g monofilament applied by a trained

diabetologist. History of macrovascular damage, namely cerebrovascular disease (CVD), coronary heart disease (CHD) and peripheral artery disease (PAD) was reported. CVD was defined as a history of stroke or transient ischemic attack. CHD was defined as a history of myocardial infarction, angina pectoris, coronary artery angioplasty or bypass. PAD was defined as clinical signs combined with pathological findings on duplex ultrasound or computed tomography in lower limbs, or as a history of lower limb angioplasty or surgical revascularization.

## Statistics

Continuous variables are expressed as median (25th–75th percentile) for population description, or mean +standard error of the mean for *in vitro* experiments (Prism software v9.1.2). Categorical variables were described as percentages. T2D patients and controls baseline data were compared with Wilcoxon signed-rank test, or Fisher's exact test, and a multivariate logistic regression analysis was performed with adjustment for age, sex, obesity, dyslipidemia and HTN status. Biomarker distributions before and after ultracentrifugation were compared using the Wilcoxon matched-pairs signed-rank test. Correlations between biomarker levels and conventional clinical or biological parameters were tested by Spearman test. Biomarkers levels in specific T2D patient subgroups were compared by Wilcoxon signed-rank test. Potential association of log-transformed plasma absorbance with cardiovascular risk factors, vascular complications and treatments in T2D patients were assessed using multivariate logistic regressions adjusted for age and sex. Missing data were considered as negligible (<9%), except for MCV (31%), MCHC (31%), fasting glycemia (66%). Only a few datapoints could be collected for bilirubin (7%) and LDH (6%) levels. Missing data were not imputed. Statistical analyses were two-sided and performed with the R software v3.6.2 (R Foundation for Statistical Computing, Vienna, Austria). Statistical significance was defined as $p < 0.05$.

## Results

We quantified direct biomarkers of IVH in 109 T2D patients, focusing on RBC contents (Hb, heme) and membranes (EV).

Heme-related absorbance levels (Abs398) were increased by 57% in T2D patients vs. controls (0.508 vs. 0.324, $p < 0.001$ respectively) (Fig 1A) and Hb-related absorbance (Abs575) levels by 60% (0.040 vs. 0.025, $p = 0.003$ respectively) (Fig 1B) in univariate analyses. Abs398 remained associated to T2D after baseline biological data were adjusted for age, sex and the presence of obesity, HTN and dyslipidemia ($p = 0.020$), criteria which differed slightly in T2D patients from controls (S1 Table).

Large EV were quantified in plasma by conventional flow cytometry (FACS), after labeling surface PS and RBC-specific CD235a (Fig 1C and 1D). EV levels, RBC-derived or PS+, did not differ between T2D and controls ($p > 0.05$), but measurements were more dispersed in T2D vs. controls.

We characterized EV from T2D and control plasma by CryoTEM to avoid potential artifacts due to dyslipidemia in T2D patients. EV were identified as circular objects between 40 and 3000 nm in diameter, with electron-dense membranes (Fig 1E and 1F). Lipoproteins and other objects which displayed no extracellular membrane were ignored. Less than 0.5% EV were observed above 1200 nm. EV were 17% smaller in T2D vs. control plasma (158+/-4 vs. 189+/-6 nm; $p = 0.011$), with a higher prevalence of small EV below 100 nm (21.3% in T2D vs. 13.4% in controls).

We assessed the proportion of EV-bound vs. 'free' heme-related absorbance in plasma. We repeated measurements in intact plasma and plasma depleted of EV by ultracentrifugation. Abs398 was reduced by 41% in EV-depleted control plasma and by 54% in EV-depleted T2D

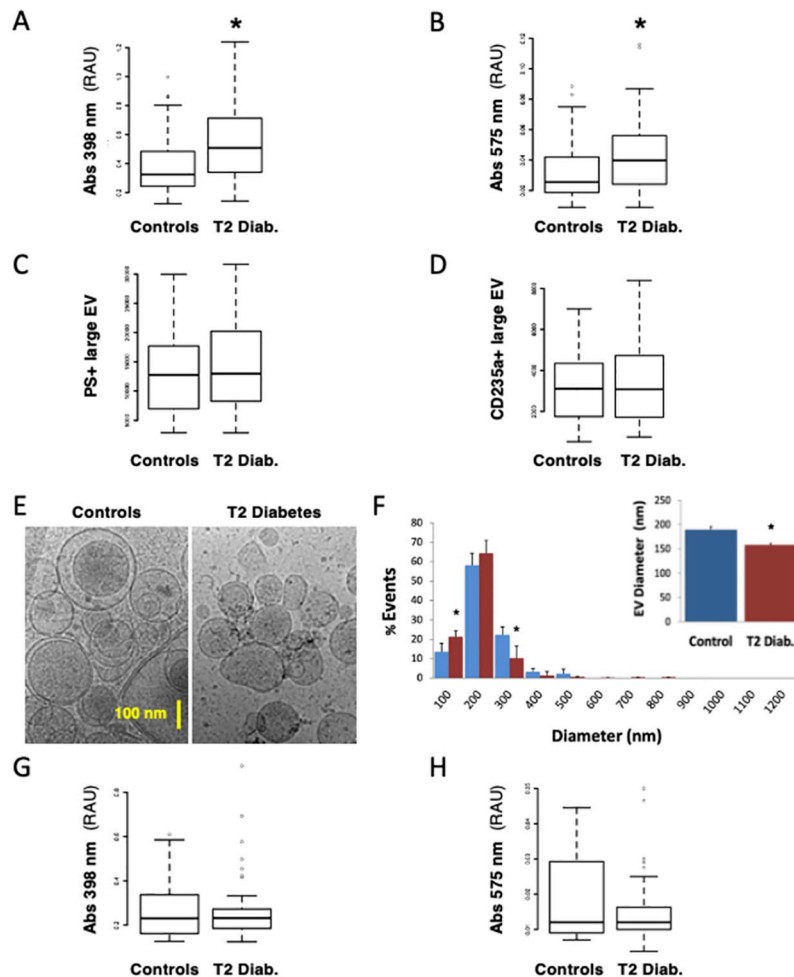

**Fig 1. Intravascular hemolysis in patients with type 2 diabetes vs. control subjects.** We quantified direct biomarkers of intravascular hemolysis (IVH) in 109 patients with type 2 diabetes (T2D) vs. 65 non-T2D control subjects, focusing on RBC contents and extracellular vesicles (EV). RBC contents and RBC membrane EV were quantified in platelet-free plasma of control (Blue) vs. T2D patients (Red). Heme-related absorbance was measured at Soret band peaks at 398 nm **(A)** and 575 nm **(B)**. Circulating large EV were measured by flow-assisted cell sorting after labeling of surface PS with fluorescent annexin-A5 **(C)** or a RBC-specific anti-CD235a (glycophorin-A) antibody **(D)**. EV from platelet-free plasma were analyzed by cryogenic, transmission electron microscopy **(E)**. EV were identified as objects between 40 and 1200 nm in diameter, with electron-dense membranes and circularity. EV diameters were extracted and presented as size-distribution by 100 nm increments, or as average size **(F)**. Absorbance associated to EV was deduced from signals remaining after depleting EV by ultracentrifugation (25000 g, 4 h) **(G-H)**. (*) p<0.05 vs. Controls.

plasma (*p* = 0.72 vs. controls) (Fig 1G). Abs575 was reduced by 53% in controls and 62% in T2D (*p* = 0.32 vs. controls) (Fig 1H).

Next, we aimed to characterize differences in the phenotype and function of EV released by RBC from control and T2D patients *in vitro*. RBC were subjected to $Ca^{2+}$ influx or shear stress and supernatants were analyzed by CryoTEM (Fig 2A–2E). EV from T2D RBC were 12% smaller in diameter than control EV after $Ca^{2+}$ influx (154+/-2 vs. 175+/-3 nm; *p*<0.004), and 37% smaller after shear stress (136+/-2 vs. 217+/-8 nm; *p*<0.002). There was a higher prevalence of small EV shed by T2D RBC after $Ca^{2+}$ influx (15.6% vs. 8.8% for controls) and shear stress (49.5% vs. 19.5% for controls). NTA confirmed CryoTEM results obtained on RBC-derived EV (Fig 2F–2H), with 8% to 20% reduction in the diameters of EV derived from T2D

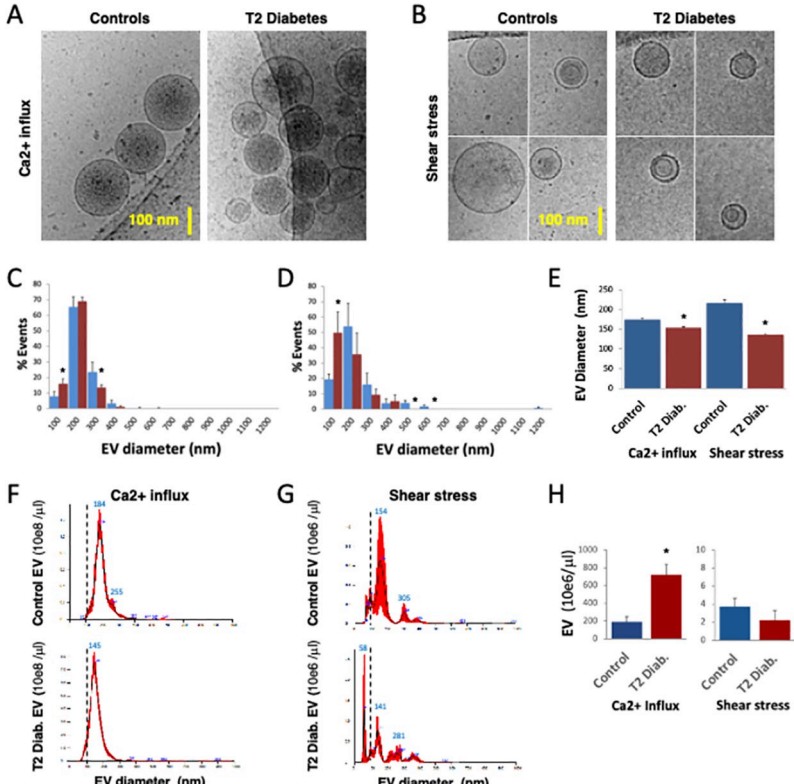

**Fig 2. RBC from patients with type 2 diabetes release abnormal EV.** Purified RBC from control (Blue) or obese T2D (Red) patient blood (n = 4) were stressed by sudden addition of $Ca^{2+}$ ionophore (A23187) **(A, C, F)**, or application of shear stress (Dounce homogenizer strokes) **(B, D, G)**. Supernatants were collected and analyzed by cryogenic, transmission electron microscopy (CryoTEM) **(A-E)**, or nanoparticle tracking analysis (NTA) **(F-H)**. EV size distributions were computed in categories of 100 nm increment **(C, D)**, and considered 'small' below 100 nm in diameter (interrupted bars). Average EV sizes were compared according to parental RBC (Controls, T2D) and type of vesiculation **(E, H)**. EV size distributions **(F, G)** and concentrations were assessed by NTA **(H)**. T2D RBC-derived EV were 12% to 37% smaller in diameter than control EV, with a 2-fold increase in the prevalence of small EV (<100 nm). NTA also showed that T2D RBC stimulated by $Ca^{2+}$ influx shed 3-fold more EV than control RBC. In general, $Ca^{2+}$ influx triggered 50 to 300-fold more EV than shear stress. (*) p<0.05 vs. Controls.

RBC. NTA analysis also revealed that $Ca^{2+}$ influx triggered 50 to 100-fold more EV than shear stress in all RBC ($p<0.05$). T2D RBC released 3-fold more EV than control RBC after sudden $Ca^{2+}$ influx ($p<0.002$) (Fig 2H).

We ultracentrifuged $Ca^{2+}$ influx-stimulated RBC supernatants, removing over 85% to 95% EV by NTA (Fig 3A). In these supernatants, Abs398 and Abs575 measurements suggested that hemoglobin release was primarily EV-independent. However, 5% to 20% of Abs398 and Abs575 remained associated to RBC EV (Fig 3B and 3C).

RBC-derived EV stimulated by $Ca^{2+}$ influx were applied to human endothelial cultures (HUVEC) for 2h (25 to 1000 10e5 EV/mL) and we measured ROS production. EV from T2D RBC triggered ROS accumulation ($p<0.05$; Fig 3D). This was inhibited by heme-neutralizing hemopexin and PS-neutralizing annexin-A5, and sensitive to protein kinase C inhibitors (Ro-31-8220 and Gö-6976) (Fig 3E).

To further demonstrate differences in the nature of EV from control and T2D RBC, we used the thrombin generation assay, which depends on the ability of membrane to recruit coagulation cascade factors. We compared thrombin generation potentials of EV from T2D or control RBC stimulated by $Ca^{2+}$ influx in vitro. EV were added (10e9 EV/mL) to platelet-free

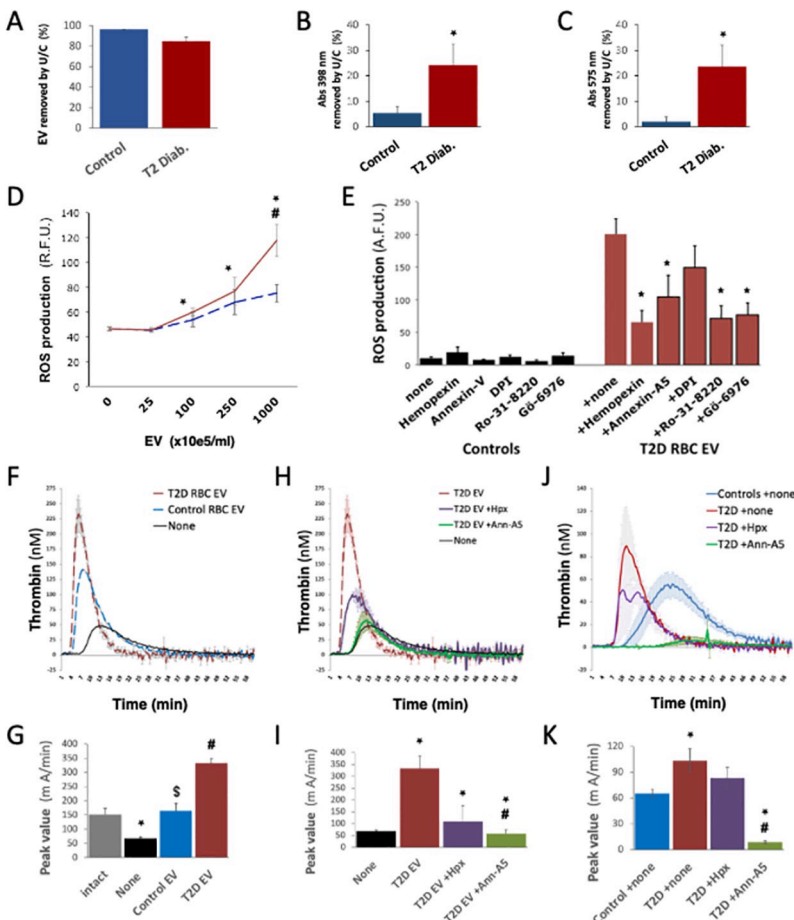

**Fig 3. EV from T2D RBC carry heme, stimulate oxidative stress and support thrombin generation.** EV shed by T2D RBC after Ca2+ influx were characterized. EV removed from RBC supernatants by ultracentrifugation were quantified by nanoparticle tracking analysis (NTA) **(A)**, and the drop in heme-related absorbance at 398 nm and 575 nm by spectrophotometry **(B-C)**. RBC EV (up to 10e8 EV/mL) were applied to cultured endothelial cells (HUVEC). Radical oxygen species production was quantified by fluorescent probe after 1 hour, in the presence of heme-antagonist hemopexin (Hpx), PS-neutralizing annexin-A5, NADPH oxidase inhibitor diphenyleneiodonium chloride (DPI), or the protein kinase-C inhibitors Ro-31-8220 and Gö-6976 **(D-E)**. RBC-derived EV triggered endothelial ROS production in a dose-dependent fashion, above 10e7 EV/mL. We used thrombin activation (CAT assay) to detect the ability of EV to support PS-mediated reactions. To focus on EV surface PS, we added control or T2 diabetic RBC EV (10e9 EV/mL) to EV-depleted, platelet-free control plasma, and implemented CAT without synthetic phospholipids **(F-I)**. Alternatively, we used platelet-free with endogenous EV, supplemented with hemopexin (Hpx) or annexin-A5 **(J-K)**. We show curves of thrombin generation over 60 minutes **(F,H,J)** and maximum peak values **(G,I,K)**. Thrombin generation was increased and accelerated in T2D vs. control plasma, in an annexin-A5-dependent fashion. (*) $p < 0.05$ vs. Controls. (#) $p < 0.05$ vs T2D EV.

and EV-depleted control plasma. Thrombin activation was triggered by recalcification (only). T2D EV supported accelerated thrombin activation than control EV, with a 2-fold increase in peak value (Fig 3F and 3G). T2D EV were partly inhibited by hemopexin and fully blocked by annexin-A5 (Fig 3H and 3I). In platelet-free plasma (ie. including endogenous EV), thrombin activation was accelerated and peak values were increased 2-fold in T2D vs. control plasma. But thrombin generation in T2D plasma could be fully inhibited by annexin-A5 and partly by hemopexin (Fig 3J and 3K).

Next, we assessed the association of our IVH biomarkers with clinical data and vascular injury in T2D patients, identifying groups with obesity, HTN and dyslipidemia. Clinical

laboratory data revealed a 3.4% drop in RBC volume (MCV) in T2D, overall (85.3 vs. 82.9 mm$^3$, $p = 0.027$), using multivariate analysis adjusted for age, sex, obesity, HTN and dyslipidemia (S1 Table). RBC volume was negatively correlated with HbA1c across the cohort ($r = -0.31$, $p = 0.004$) and among T2D patients ($r = -0.33$ $p = 0.032$). We found no association of T2D with biomarkers like RBC counts, hemoglobinemia, hematocrit, AST, ALT, LDH or iron levels. T2D associated with lower bilirubin levels vs. control patients, despite the few data collected (5.5 vs. 8.5 μmol/l, $p = 0.036$).

In T2D, we found no correlation of plasma Abs398 and Abs575 levels with HbA1c (Fig 4A and 4E), fasting glycemia, C-peptide levels or T2D duration (Table 1). However, Abs398 and Abs575 were 27% and 31% higher in obese vs. non-obese T2D patients, respectively (*OR* = 4.8;

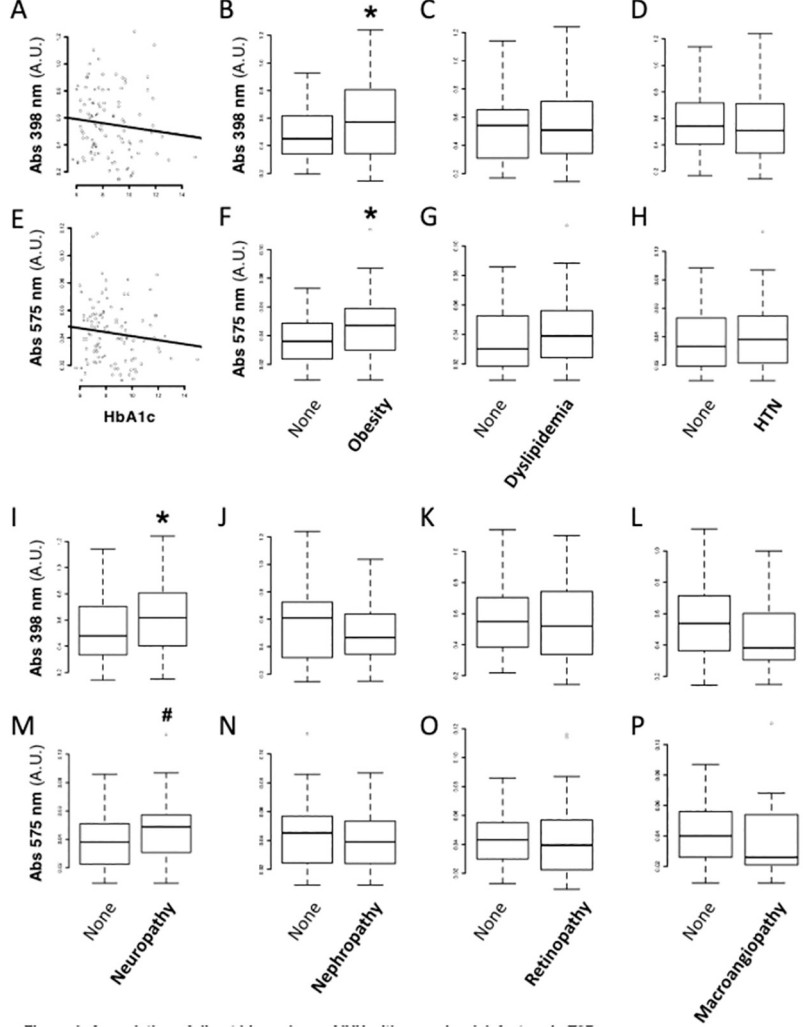

**Fig 4. Association of direct biomarkers of IVH with vascular risk factors in type 2 diabetes.** We crossed our direct IVH biomarkers with clinical data. We investigated the association of plasma absorbance at 398 nm **(A-D, I-L)** and 575 nm **(E-H, M-P)** with subgroups of patients with T2D. First, we assessed the correlation of Abs398 and Abs575 with glycated hemoglobin (Hb1Ac) levels (r = -0.12 and -0.11, respectively. p>0.2 for both). Next, we assessed their association with obesity, dyslipidemia and hypertension (HTN) **(A-H)**. Abs398 and Abs575 were significantly increased in obese vs. non-obese patients with T2D. Finally, we assessed their association with peripheral neuropathy, nephropathy, retinopathy or macroangiopathy **(I-P)**. (*) p<0.05 vs. Controls (#) p = 0.051 vs Controls. (see Table 2 for adjusted logistic regressions using Log-transformed absorbances).

**Table 1. Correlations between IVH markers and baseline data in T2D patients after adjustment for age and sex.**

| | Abs398nm | | | Abs575nm | | |
|---|---|---|---|---|---|---|
| | *n* | *R* | *p* | *n* | *R* | *p* |
| Age | 109 | 0.016 | 0.810 | 108 | 0.006 | 0.685 |
| BMI | 107 | 0.149 | 0.167 | 106 | 0.228 | 0.051 |
| HbA1c | | -0.122 | 0.289 | 105 | -0.114 | 0.220 |
| T2D duration | 108 | 0.024 | 0.514 | 107 | 0.021 | 0.857 |
| Insulin units/day | 65 | 0.252 | **0.014*** | 64 | 0.295 | **0.010*** |
| C peptid | 102 | 0.246 | 0.080 | 101 | 0.206 | 0.071 |
| Fasting glycemia | 37 | 0.178 | 0.145 | 37 | 0.061 | 0.900 |
| eGFR (CKD EPI) | 107 | -0.102 | 0.346 | 106 | -0.075 | 0.623 |
| SBP | 105 | -0.045 | 0.743 | 104 | 0.018 | 0.726 |
| DBP | 105 | -0.113 | 0.117 | 104 | 0.056 | 0.738 |
| Heart rate | 103 | 0.175 | 0.649 | 102 | 0.082 | 0.944 |
| RBC | 109 | 0.022 | 0.541 | 108 | 0.042 | 0.787 |
| Hemoglobin | 109 | -0.003 | 0.448 | 108 | -0.015 | 0.811 |
| Hematocrit | 109 | -0.035 | 0.697 | 108 | -0.001 | 0.812 |
| MCV | 75 | -0.124 | 0.252 | 74 | -0.038 | 0.798 |
| MCHC | 75 | 0.102 | 0.363 | 74 | -0.038 | 0.921 |
| Leukocytes | 109 | 0.073 | 0.797 | 108 | 0.044 | 0.812 |
| Neutrophils | 107 | 0.097 | 0.627 | 106 | 0.066 | 0.922 |
| CRP | 108 | 0.249 | 0.272 | 107 | 0.128 | 0.818 |
| Platelets | 109 | 0.117 | 0.333 | 108 | 0.101 | 0.928 |
| AST | 105 | 0.063 | 0.851 | 105 | -0.065 | 0.174 |
| ALT | 109 | 0.102 | 0.273 | 108 | 0.116 | 0.322 |
| Iron (Serum) | 103 | 0.010 | 0.900 | 102 | 0.047 | 0.567 |
| Bilirubin | 8 | -0.090 | 0.518 | 8 | -0.160 | 0.601 |
| LDH | 7 | -0.310 | 0.144 | 7 | -0.286 | **0.046*** |

*$p < 0.05$; all statistics are adjusted for age and sex (Age adjusted for sex only)

Data for bilirubin and LDH were only collected for 7% patients. BMI, body mass index; AST, aspartate aminotransferase; ALT, alanine aminotransferase; LDH, lactate dehydrogenase; CRP, C reactive protein; SBP, systolic blood pressure; DBP, diastolic blood pressure; RBC, red blood cells; MCV, mean corpuscular volume; MCHC, mean corpuscular hemoglobin concentration; eGFR, estimated glomerular filtration rate.

$p = 0.041$ and $OR = 4.6$; $p = 0.017$) (Fig 4B and 4F; Table 2), and positively associated to the total daily dose of insulin ($r = 0.25$, $p = 0.014$ and $r = 0.29$, $p = 0.010$) (Table 1).

Abs398 and Abs575 were not associated with RBC counts, hemoglobinemia, hematocrit, bilirubin, AST, ALT, LDH and iron levels, or inflammation (Leukocyte counts, CRP) (Tables 1 and 2), nor correlated to dyslipidemia or HTN (Fig 4C, 4D, 4G and 4H), lipid-lowering or antihypertensive medication (Tables 2 and 3).

We tested potential association of IVH biomarkers with overt micro- or macrovascular damage in T2D. Abs398 levels ($OR = 5.5$; $p = 0.040$) and Abs575 levels ($OR = 5.1$; $p = 0.042$) were increased in T2D patients suffering from peripheral sensory neuropathy, by 29% and 32% respectively (Fig 4I and 4M; Table 2). This unique association remained significant after adjustment for factors classically involved in the pathogenesis of T2D neuropathy, such as HbA1c, T2D duration, age and sex (S2 Table). Abs398 or Abs575 were unrelated to nephropathy, retinopathy or macrovascular injury ($p > 0.3$) (Fig 4J–4L and 4N–4P; Table 2).

Finally, circulating levels of PS+ and CD235a+ large EV were not statistically elevated in T2D, and large EV did not correlate with the circulating levels of HbA1c ($R = -0.384$; $p > 0.055$

**Table 2. Associations between IVH biomarkers and cardiovascular risk factors, or micro- and macrovascular complications in T2D patients: Odds Ratios and their 95% confidence intervals after adjustment for age and sex.**

| | Abs398 | | | | | Abs575 | | | | |
|---|---|---|---|---|---|---|---|---|---|---|
| | No | Yes | n | OR (logAbs398) | p | No | Yes | n | OR (logAbs575) | p |
| Male sex | 0.519 (0.336–0.713) | 0.505 (0.377–0.725) | 109 | 1.6 (0.27;10.11) | 0.609 | 0.043 (0.025–0.056) | 0.039 (0.024–0.054) | 108 | 0.80 (0.17;3.75) | 0.771 |
| Hypertension | 0.542 (0.405–0.716) | 0.507 (0.34–0.709) | 108 | 0.52 (0.04;5.47) | 0.594 | 0.043 (0.032–0.056) | 0.039 (0.023–0.055) | 107 | 1.01 (0.12;7.98) | 0.995 |
| Dyslipidemia | 0.540 (0.652;0.308) | 0.507 (0.342–0.711) | 108 | 2.04 (0.19;21.04) | 0.546 | 0.044 (0.025–0.053) | 0.039 (0.024–0.056) | 107 | 1.59 (0.20;12.24) | 0.654 |
| **Obesity** | 0.450 (0.341–0.617) | 0.571 (0.357–0.791) | 107 | 4.82 (0.59;42.52) | **0.041**[*] | 0.036 (0.024–0.049) | 0.047 (0.03–0.059) | 106 | 4.62 (0.76;31) | **0.017**[*] |
| Tobacco use | 0.540 (0.339–0.714) | 0.411 (0.381–0.504) | 108 | 0.32 (0.01;8.11) | 0.483 | 0.041 (0.024–0.056) | 0.034 (0.024–0.044) | 107 | 0.45 (0.03;6.65) | 0.553 |
| Microangiopathy | 0.610 (0.359–0.708) | 0.507 (0.343–0.735) | 108 | 0.41 (0.05;3.12) | 0.400 | 0.047 (0.030–0.056) | 0.039 (0.023–0.056) | 107 | 0.49 (0.08;2.77) | 0.428 |
| **Neuropathy** | 0.478 (0.334–0.699) | 0.617 (0.404–0.808) | 106 | 5.54 (1.01;40.12) | **0.040**[*] | 0.038 (0.022–0.050) | 0.049 (0.031–0.057) | 105 | 5.10 (1.01;29.7) | **0.042**[*] |
| Nephropathy | 0.610 (0.326–0.719) | 0.466 (0.345–0.517) | 107 | 0.58 (0.08;3.88) | 0.576 | 0.045 (0.024–0.057) | 0.039 (0.024–0.053) | 106 | 0.81 (0.15;4.26) | 0.806 |
| Microalbuminuria | 0.53 (0.332–0.735) | 0.465 (0.394–0.589) | 102 | 1.02 (0.13;8.39) | 0.988 | 0.043 (0.023–0.057) | 0.039 (0.029–0.048) | 101 | 1.19 (0.20;7.42) | 0.849 |
| Proteinuria | 0.61 (0.338–0.808) | 0.451 (0.347–0.589) | 101 | 0.295 (0.04;1.98) | 0.215 | 0.047 (0.027–0.059) | 0.038 (0.023–0.051) | 100 | 0.46 (0.09;2.31) | 0.348 |
| eGFR <60mL/min | 0.504 (0.341–0.544) | 0.526 (0.369–0.726) | 107 | 1.63 (0.19;14.69) | 0.654 | 0.039 (0.024–0.055) | 0.044 (0.026–0.058) | 106 | 2.29 (0.37;15.57) | 0.378 |
| Retinopathy | 0.549 (0.386–0.702) | 0.526 (0.344–0.791) | 104 | 0.61 (0.09–3.89) | 0.605 | 0.043 (0.030–0.055) | 0.039 (0.022–0.057) | 103 | 0.522 (0.10;2.55) | 0.427 |
| Maculopathy | 0.543 (0.383–0.733) | 0.463 (0.395–0.572) | 99 | 0.44 (0.02;10.34) | 0.599 | 0.042 (0.027–0.056) | 0.039 (0.032–0.056) | 98 | 1.18 (0.09;19.6) | 0.902 |
| Laser for retinopathy | 0.541 (0.344–0.715) | 0.485 (0.414–0.769) | 104 | 2.20 (0.25;23.21) | 0.491 | 0.042 (0.024–0.056) | 0.039 (0.031–0.057) | 103 | 2.11 (0.31;16.88) | 0.456 |
| Cardiovascular disease | 0.537 (0.369–0.712) | 0.401 (0.306–0.682) | 108 | 0.37 (0.03;4.17) | 0.418 | 0.04 (0.026–0.056) | 0.026 (0.021–0.054) | 107 | 0.33 (0.04;2.69) | 0.296 |
| Coronary heart disease | 0.526 (0.342–0.711) | 0.401 (0.264–0.583) | 108 | 0.18 (0;5.06) | 0.301 | 0.04 (0.025–0.056) | 0.027 (0.019–0.052) | 107 | 0.25 (0;4.57) | 0.342 |
| Stroke | 0.513 (0.342–0.711) | 0.444 (0.308–0.653) | 108 | 0.28 (0–8.62) | 0.451 | 0.040 (0.025–0.056) | 0.037 (0.024–0.058) | 107 | 0.54 (0.03;10.38) | 0.665 |
| Peripheral arterial disease | 0.526 (0.347–0.714) | 0.38 (0.325–0.516) | 108 | 0.01 (0–5.1) | 0.292 | 0.04 (0.025–0.056) | 0.029 (0.02–0.045) | 107 | 1.85 (0;3.62) | 0.271 |

[*]$p < 0.05$; all statistics are adjusted for age and sex (Sex adjusted for age only). Data are median ($25^{th}$–$75^{th}$ percentiles). Obesity, BMI$\geq$30kg/m²; eGFR, estimated glomerular filtration rate.

after adjustment for age and sex). PS+ and CD235a+ large EV were not associated with dyslipidemia, HTN or obesity as cardiovascular risk factors, nor with vascular damage in T2D patients (S3 Table).

## Discussion

We demonstrated for the first time that T2D associates with IVH, and described new circulating biomarkers of peripheral sensory neuropathy. IVH measurements in T2D plasma formed a complex picture: Previous reports of elevated levels of large, RBC-derived EV, analyzed by

**Table 3. Associations between IVH markers and treatments in T2D patients: Odds Ratios and their 95% confidence intervals after adjustment for age and sex.**

| | Abs398 | | | | | Abs575 | | | | |
|---|---|---|---|---|---|---|---|---|---|---|
| | No | Yes | n | OR (logAbs398) | p | No | Yes | n | OR (logAbs575) | p |
| Insulin | 0.497 (0.341–0.706) | 0.534 (0.364–0.744) | 107 | 1.67 (0.25;11) | 0.592 | 0.040 (0.026–0.053) | 0.041 (0.024–0.058) | 106 | 1.75 (0.34;9.05) | 0.498 |
| Metformin | 0.543 (0.37–0.812) | 0.504 (0.342–0.706) | 107 | 0.47 (0.06;3.55) | 0.474 | 0.039 (0.024–0.069) | 0.040 (0.025–0.054) | 106 | 0.47 (0.08;2.68) | 0.409 |
| Sulfonylurea | 0.526 (0.351–0.808) | 0.490 (0.343–0.652) | 107 | 0.38 (0.06;2.34) | 0.298 | 0.039 (0.023–0.061) | 0.041 (0.026–0.053) | 106 | 0.73 (0.15;3.53) | 0.696 |
| DPP-4 inhibitor | 0.563 (0.369–0.791) | 0.452 (0.28–0.543) | 107 | 0.14 (0.02;1.10) | 0.064 | 0.047 (0.028–0.057) | 0.032 (0.017–0.044) | 106 | 0.14 (0.02;0.8) | **0.030**\* |
| GLP-1 analog | 0.526 (0.341–0.714) | 0.463 (0.398–0.713) | 107 | 0.736 (0.03;19.84) | 0.848 | 0.04 (0.024–0.056) | 0.039 (0.030–0.055) | 106 | 1.03 (0.06;21.62) | 0.986 |
| ACEi | 0.541 (0.397–0.721) | 0.384 (0.307–0.641) | 107 | 0.23 (0.03;1.77) | 0.160 | 0.042 (0.026–0.057) | 0.031 (0.022–0.051) | 106 | 0.23 (0.04;1.35) | 0.104 |
| ARB | 0.497 (0.309–0.709) | 0.543 (0.405–0.725) | 107 | 4.18 (0.52;38.61) | 0.189 | 0.039 (0.022–0.054) | 0.047 (0.031–0.057) | 106 | 6.01 (0.93;46.64) | 0.070 |
| Diuretic | 0.505 (0.344–0.707) | 0.519 (0.333–0.713) | 107 | 1.08 (0.16;7.17) | 0.939 | 0.04 (0.023–0.053) | 0.039 (0.024–0.058) | 106 | 1.84 (0.36;10.03) | 0.467 |
| Calcium Chanel Blocker | 0.537 (0.353–0.710) | 0.461 (0.336–0.732) | 107 | 0.61 (0.09;4.07) | 0.608 | 0.04 (0.025–0.56) | 0.039 (0.023–0.056) | 106 | 0.58 (0.11;3.04) | 0.516 |
| Anti-platelet agent | 0.504 (0.349–0.656) | 0.598 (0.329–0.821) | 107 | 2.44 (0.30;22.08) | 0.412 | 0.039 (0.025–0.052) | 0.048 (0.022–0.067) | 106 | 1.54 (0.02–10.23) | 0.647 |
| Aspirin | 0.504 (0.342–0.654) | 0.627 (0.337–0.822) | 107 | 4.40 (0.54;42.14) | 0.179 | 0.039 (0.025–0.053) | 0.049 (0.023–0.067) | 106 | 2.56 (0.41;17.82) | 0.322 |
| Lipid lowering therapy | 0.485 (0.372–0.645) | 0.534 (0.336–0.809) | 107 | 1.99 (0.28;14.36) | 0.491 | 0.04 (0.03–0.053) | 0.039 (0.023–0.056) | 106 | 1.07 (0.19;6.02) | 0.934 |
| Statin | 0.464 (0.359–0.631) | 0.542 (0.337–0.809) | 107 | 2.78 (0.42;19.25) | 0.291 | 0.039 (0.029–0.053) | 0.040 (0.023–0.057) | 106 | 1.44 (0.27;7.63) | 0.665 |

\*$p < 0.05$; all statistics are adjusted for age and sex. Data are median (25th–75th percentiles)

DPP4, dipeptidyl peptidase-4; GLP1, Glucagon-like peptide-1; ACEi, angiotensin converting enzyme inhibitor; ARB, angiotensin II receptor blocker.

FACS, remain conflicting. Here, a possible increase in large RBC-derived EV levels detected by FACS in T2D patient vs. control plasma did not reach significance.

Conversely, RBC volume (MCV) was reduced by 3.4% in T2D, reflecting either RBC dehydration or EV shedding. Reduced MCV was combined with an abnormal leak of heme into the T2D circulation (about 50% above controls, and about 30% higher in obese vs. non-obese T2D patients). The reduced RBC volume was inversely correlated with circulating HbA1c levels, while heme-related absorbance in plasma was positively correlated to daily insulin requirements. In summary, RBC injury and RBC leakage into the circulation imply that RBC breakdown occurs in T2D, that T2D patients suffer from underestimated, low grade IVH, and that the phenomenon may be linked to glycemia. However, we found no link between heme-related absorbance and glycemia, c-peptide levels, or T2D duration. We concluded that RBC damage and IVH can emerge once T2D is established, but RBC and Hb glycation alone may not be sufficient to cause IVH.

In T2D plasma, depleting 80% to 95% EV by centrifugation reduced heme-related absorbance by 50%, a proportion similar to hemolytic disorders [12]. In stressed RBC supernatants, the proportion of EV-associated, heme-related absorbance was close to 20%. Moreover, EV from T2D RBC triggered ROS production in human endothelial cultures and enhanced thrombin activation in a heme- and PS–mediated fashion. This demonstrated that EV released

by T2D RBC differ from those shed by control RBC in volume, cargo (pro-oxidant Hb and heme degradation products) and membrane surface properties. RBC-derived EV may thus enhance vascular degeneration and inflammation in T2D, as they do in hemolytic diseases [12, 31]. RBC-derived EV may also favor coagulopathy in T2D, an aspect that deserves a dedicated study.

In vitro, RBC isolated from T2D patients shed about 3-fold more EV than control RBC under similar stimulation (sudden $Ca^{2+}$ influx or shear stress), echoing previous reports that RBC from obese T2D patients release more EV than other RBC [14–17]. CryoTEM and NTA studies also showed that EV produced by isolated T2D RBC were 20% to 40% smaller in diameter than control EV. Circulating EV were also about 20% smaller in T2D patients vs. control plasma. This implies that the smaller size of T2D EV resulted from acquired modifications of parent blood cells, modulating vesiculation and EV characteristics. RBC glycation may weaken T2D RBC integrity and translate into EV shedding. However, an additional stimulus linked to obesity, like inflammation, oxidative stress or shear stress, may synergize with RBC remodeling to trigger vesiculation and IVH in T2D [41].

CryoTEM and NTA revealed that the prevalence of small EV below 100 nm was increased by 60% to 100% in T2D. Van der Pol, Nieuwland and colleagues [42] established that conventional FACS technology can only detect EV accurately above 300–600 nm in diameter. Hence, a drop from 190 nm to 158 nm in average EV diameter, with twice more EV of small size (80 nm average), invisible by FACS, must impact EV quantification in T2D. Furthermore, routine evaluation of plasma EV by FACS remains challenging for clinical biology platforms, regarding equipment availability and procedure standardization [11, 33, 42]. The phenotype of T2D EV highlights the weakness of conventional FACS, supports the need for multiple approaches to characterize EV, and reduces the relevance of conventional FACS to develop plasma EV as biomarkers in cardiovascular disease. Unfortunately, cryoTEM is low-throughput, qualitative only, and maladapted to patient screening, but it proved irreplaceable to unravel the smaller size of EV in T2D.

Very few circulating biomarkers apart from glycated Hb (HbA1c) and advanced glycation endproducts (AGE) [43] can help stratify T2D patients at risk of microvascular injury in clinical practice. Current clinical tests used to detect early vascular complications in T2D require expertise, they are costly and sometimes unavailable. Our new, direct biomarkers of IVH, like plasma heme-related absorbance, may be more practical and less costly to screen for T2D microangiopathy. In our T2D cohort, heme-related absorbance was not a general marker of vascular complication, it was specifically associated with peripheral sensory neuropathy. This might relate to a higher sensitivity of peripheral neurons to IVH products, or to unique characteristics of the vascular bed and hemorheology in the lower leg. However, peripheral neuropathy is a multifactorial complication, which may integrate microvascular injury and nerve inflammation.

Diabelyse was a cross-sectional, observational study with 174 patients. Larger, multicentric, prospective studies with higher statistical power are now needed to clarify the association of heme-related absorbance with microvascular complications in T2D, and their relevance to stratify T2D patients at risk of developing peripheral neuropathy. Neuropathy remains a major debilitating complication of T2D, with risk of foot injury, neuropathic pain and autonomic failure, in millions of patients with T2D across the world [44]. Absorbance is technically simple to measure, spectrophotometers are widely available and standardization is straightforward. Heme-related absorbance may ultimately, represent a first line method to screen patients susceptible to neuropathy for an earlier diagnostic. Patients with high levels of heme-related absorbance may eventually be prioritized for neuroprotective prophylaxis and therapeutics. Anti-hemolysis therapies, such as heme-chelating recombinant hemopexin or EV-neutralizing

annexin-A5 may also be tested to treat T2D patients with low-grade IVH or established neuropathy.

Limited regenerative anemia was previously reported in T2D, independent of iron stores, inflammation, or renal dysfunction [45, 46]. Here, a slight trend for hemolytic anemia in T2D patients was too mild to emerge from routine hospital analyses based on indirect biomarkers of hemolysis (AST, iron, bilirubin and LDH), known to be impacted by the many metabolic modifications that prevail in T2D. For instance, plasma bilirubin levels may rise with IVH, but bilirubin is known to decrease in T2D [47, 48], which we confirmed despite the very few data-points collected. Lower circulating bilirubin levels in T2D were previously attributed to a decreased HO-1 expression in the late stage of T2D, limiting heme catabolism [48]. Heme-derived degradation species displaying Soret band absorbance may be comparatively enriched in T2D and obese T2D plasma. This makes plasma spectrophotometry a candidate for a cost-efficient, reproducible and high-throughput biomarker.

The pathological consequences of IVH are mainly due to the production and actions of oxidized Hb forms. After release in the bloodstream, Hb oxidates and generates different products, including metHb, ferrylHb and covalently crosslinked Hb forms. MetHb mainly issues from the reaction between oxy-Hb and NO, or Hb auto-oxidation. FerrylHb is a transient compound produced by Hb or metHb oxidation by peroxides (hydrogen peroxide or lipid hydroperoxides). FerrylHb can oxidize aminoacids residues near b-globin chains, leading to the production of globin-derived radicals, which enhance covently crosslinked Hb multimerization [49].

These products of Hb oxidation can stimulate specific pro-oxidant and pro-inflammatory actions on the vascular endothelium, with functional consequences. MetHb triggers the endothelial production of interleukin-6 and -8. FerrylHb can trigger endothelial activation especially through Nuclear Factor-kappa B (NFkB) pathway. Globin-derived radicals activate endothelial interleukin-1ß production via the NOD-like receptor family, pyrin domain containing 3 (NLRP3) inflammasome [49]. In addition, oxidized forms of Hb shed their prosthetic heme more readily. Some suppose that protein-free heme could trigger specific oxidative reactions in the bloodstream, like hydroxyl radical formation through Fenton reaction or lipid per-oxidation [49]. However, opportunities for heme to maintain itself in a 'free' state are unlikely and would be extremely brief. The odds are that heme will rapidly be exchanged and interact with proteins present in plasma at high concentrations, like lipoproteins, hemopexin or albumin.

It would be advantageous to know which forms of oxidized Hb and derivatives prevail in T2D plasma in order to imagine therapeutic strategies for their neutralization. Future studies could attempt to distinguish Hb oxidation products in T2D, possibly via absorbance spectroscopy with spectral deconvolution techniques [26, 50] or via reverse-phase high performance liquid chromatography (RP-HPLC) [34] and quantify their relative proportions.

## Conclusions

Low grade IVH occurs in T2D complicated with obesity, and in non-obese patients with T2D to a lesser extent. T2D RBC have a greater potential to release heme-loaded EV of abnormally small size. Circulating EV in T2D plasma also display an abnormal small size, making classical FACS less appropriate to develop RBC-derived EV as biomarkers in T2D. Plasma levels of heme-related absorbance were independently associated with a specific microvascular complication, peripheral sensory neuropathy. Spectrophotometric markers of IVH might eventually help stratify the risk of peripheral nerve damage during T2D, and identify patients who could benefit from earlier prophylaxis and IVH-targeted personalized therapy.

## Supporting information

**S1 File. Supplementary materials & methods.** Reagents and assays, T2D cohort 'Diabelyse', Inclusion parameters, Non-inclusion parameters, Blood sample collection, Flow cytometry for extracellular vesicles.
(DOCX)

**S2 File.**
(PDF)

**S3 File.**
(DOCX)

**S4 File.**
(PDF)

**S1 Table. Baseline clinical and biological characteristics of patients included in the Diabelyse study.** The table lists the clinical data that were significantly associated with T2D in cohort Diabelyse, in univariate and multivariate analyses. Those included classical traits of T2D like high Hb1Ac, high fasting glycemia, lower bilirubin and higher leukocyte levels, but also our plasma marker of IVH, Abs398 (related to heme). In mutivariate analyses$^{\$}$, Abs575 was not significant, suggesting that Abs398 may be a more robust biomarker. The average RBC volume was also reduced in T2D. Data are median (25th–75th percentiles) or n (%). $^{*}P<0.05$. $^{\$}$ After adjustment for age, sex, obesity, dyslipidemia and HTN status. Data for bilirubin and LDH were only collected for 7% patients. BMI, body mass index; Obesity, BMI $\geq$30kg/m$^2$; eGFR, estimated glomerular filtration rate; AST, aspartate aminotransferase; ALT, alanine aminotransferase; RBC, red blood cells; MCV, mean corpuscular volume; MCHC, mean corpuscular hemoglobin concentration; LDH, lactate dehydrogenase; CRP, C reactive protein.
(DOCX)

**S2 Table. Variables associated to Diabetic Peripheral Sensory Neuropathy.** The table reveals that only two independent variables were significantly associated to Diabetic Peripheral Sensory Neuropathy in cohort Diabelyse, namely Abs398 and Abs575. In contrast, other parameters related to RBC did not, including RBC counts, serum iron, plasma bilirubin, or classical risk factors. Data are median (25th–75th percentiles) or n (%). $^{*}p<0.05$. $^{\$}$ after adjustment with factors classically involved in the pathogenesis of neuropathy in T2D: HbA1C, T2D duration, age and sex. Data for bilirubin and LDH were only collected for 7% patients. Obesity, BMI$\geq$30kg/m$^2$; eGFR, estimated glomerular filtration rate; RBC, red blood cells; MCV, mean corpuscular volume; MCHC, mean corpuscular hemoglobin concentration; LDH, lactate dehydrogenase; CRP, C reactive protein.
(DOCX)

**S3 Table. Associations of PS+ and CD235a+ large EV with cardiovascular risk factors, or micro- and macrovascular complications in T2D patients.** The table reveals that the levels of PS+ CD235+ EV were not significantly associated to any of the markers of vascular injury in T2D cohort Diabelyse (all OR >0.98; all $p$>0.065). Data are median (25th–75th percentiles), or Odds Ratios and their 95% confidence intervals after adjustment for age and sex. Obesity, BMI$\geq$30kg/m$^2$; eGFR, estimated glomerular filtration rate. PS, phosphatidylserine; EV, extracellular vesicles; CD235 is also glycophorin-235.
(DOCX)

**S1 Checklist. STROBE statement—Checklist of items that should be included in reports of observational studies.**
(DOCX)

**S2 Checklist. Plos one clinical studies checklist.**
(DOCX)

## Author Contributions

**Conceptualization:** Hélène Bihan, Olivier P. Blanc-Brude.

**Data curation:** Sylvain Le Jeune, Sihem Sadoudi, Dominique Charue, Salwa Abid, Olivier P. Blanc-Brude.

**Formal analysis:** Sylvain Le Jeune, Sihem Sadoudi, Dominique Charue, Salwa Abid, Olivier P. Blanc-Brude.

**Funding acquisition:** Olivier P. Blanc-Brude.

**Investigation:** Sylvain Le Jeune, Sihem Sadoudi, Dominique Charue, Salwa Abid, Hélène Bihan, Camille Baudry, Hélène Lelong, Tristan Mirault, Robin Dhote, Chantal M. Boulanger, Olivier P. Blanc-Brude.

**Methodology:** Jean-Michel Guigner, Dominique Helley, Hélène Bihan, Camille Baudry, Hélène Lelong, Tristan Mirault, Eric Vicaut, Jean-Jacques Mourad, Chantal M. Boulanger.

**Project administration:** Olivier P. Blanc-Brude.

**Resources:** Dominique Helley, Hélène Bihan, Camille Baudry, Hélène Lelong, Eric Vicaut, Robin Dhote, Jean-Jacques Mourad.

**Visualization:** Jean-Michel Guigner.

**Writing – original draft:** Sylvain Le Jeune, Olivier P. Blanc-Brude.

**Writing – review & editing:** Sylvain Le Jeune, Dominique Helley, Hélène Bihan, Chantal M. Boulanger, Olivier P. Blanc-Brude.

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
