## [Decision Letter · Decision Letter 0]

1 Aug 2022

PONE-D-22-13587Low Grade Intravascular Hemolysis Associates with Peripheral Nerve Injury in Type 2 DiabetesPLOS ONE

Dear Dr. BLANC-BRUDE,

Thank you for submitting your manuscript to PLOS ONE. After careful consideration, we feel that it has merit but does not fully meet PLOS ONE’s publication criteria as it currently stands. Therefore, we invite you to submit a revised version of the manuscript that addresses the points raised during the review process.

We look forward to receiving your revised manuscript.

Kind regards,

Alok Raghav, PhD

Academic Editor

PLOS ONE

Journal Requirements:

“Project ‘RépaDia’ of the Fondation de France to OBB, Project ‘BiFace’ of the Fondation pour la Recherche Médicale to OBB, project ‘Cardannex’ of Inserm Transfert to OBB.”

Reviewers' comments:

Reviewer's Responses to Questions

**Comments to the Author**

1. Is the manuscript technically sound, and do the data support the conclusions?

Reviewer #1: Yes

Reviewer #2: Yes

2. Has the statistical analysis been performed appropriately and rigorously? 

Reviewer #1: Yes

Reviewer #2: Yes

3. Have the authors made all data underlying the findings in their manuscript fully available?

Reviewer #1: Yes

Reviewer #2: Yes

4. Is the manuscript presented in an intelligible fashion and written in standard English?

Reviewer #1: Yes

Reviewer #2: Yes

5. Review Comments to the Author

Reviewer #1: The manuscript is well written and authors have focused on important topic of peripheral nerve injury in type 2 diabetes patients. The authors have concluded T2D was associated with low-grade IVH. Plasma absorbance may constitute a novel biomarker of peripheral neuropathy in T2D, while flow cytometry focusing on large EV may be maladapted to characterize RBC EV in T2D. Moreover, therapeutics limiting IVH or neutralizing RBC breakdown products might bolster

vasculoprotection in T2D. Spectrophotometric markers of IVH might eventually help stratify the risk of peripheral nerve damage during T2D, and identify patients who could benefit from earlier prophylaxis and IVH-targeted personalized therapy. The findings made by the authors are novel and scientifically sound.

Reviewer #2: In this manuscript Sylvain Le Jeune et al. investigated whether Type2 diabetes (T2D) promotes intravascular hemolysis (IVH). Spectrophotometric analysis of T2D plasma revealed that low grade IVH occurs in T2D, especially in T2D obese patients. They found that red blood cells (RBCs) from T2D patients have a greater potential to release heme-loaded extracellular vesicles (EVs). Circulating EVs in T2D plasma are smaller than EVs from control subjects. Because of abnormal small size of EVs, classical FACS is less appropriate to measure them in T2D plasma. They found that heme-related absorbance of T2D plasma is independently associated with peripheral sensory neuropathy, a specific microvascular complication of T2D. The authors showed that T2D RBC-derived EV triggers production of reactive oxygen species (ROS) in endothelial cells and thrombin activation in a phosphatidylserine- and heme-dependent manner.

T2D is a global health burden with huge impact. It is long known that T2D modifies hemoglobin (Hb), red blood cell (RBC) deformability and impairs hemorheology. RBC lysis is an etiopathogenic factor in many hemolytic diseases, mainly due to the pro-oxidant and pro-inflammatory actions of hemoglobin oxidation and break-down products. But the involvement of RBC lysis and Hb-derived species in T2D remained mainly unexplored.

Therefore this study is highly relevant, well-designed and provide new information.

I have only one concern: in the methods the authors write the following: „We did not attempt to specify the different forms of ferrous, ferryl and deoxy-Hb which are all expected to contribute to Abs575.” Pathological consequences of IVH are maily due to the production and actions of oxidized Hb forms. I am missing the description of Hb oxidation, and the specific pro-oxidant and pro-inflammatory actions of the different products (metHb, ferrylHb, covalently crosslinked Hb forms, free heme). I understand that measuring the concentration of these products are quite challenging and it was not the goal of the current work, but this should be discussed properly to encourage further work in this field.

6. PLOS authors have the option to publish the peer review history of their article (what does this mean?). If published, this will include your full peer review and any attached files.

Reviewer #1: No

Reviewer #2: **Yes: **Viktória Jeney

---

## [Author Response · Author response to Decision Letter 0]

13 Sep 2022

First, we’d like to thank the reviewers for their very positive appreciation of the manuscript. 

In the 1st round of revision, the editors recommended some style modifications, some clarifications, and one of Reviewer #2 had a single request, for added discussion on a specific topic. All points were addressed and responses are provided below in extenso: 

Journal / Editors requirements 

1. File naming: 

We ensured that the manuscript sticks more closely to the PLOS ONE style, and modified some of the section, figure and table titles accordingly, the captions and citing styles, throughout the document. 

2. Participant consent: 

We provide the mentioned additional details regarding participant consent. We checked that the phrase ‘written consent’ is indeed present in extenso in the text. We regrouped two sentences into one more definitive statement. Please, see Page 4 in last paragraph. 

3. Data not shown: 

We have removed the phrase ‘data not shown’ from the manuscript. On one hand, we inserted actual data in text (Page 12, top paragraph, as below). On the other hand, we inserted a new table (S3 Table) gathering all the statistics linked to the absence of association of circulating levels of vesicles with vascular injury markers in T2D patients, to show the actual data. Please see Results, Page 12: 

“Finally, circulating levels of PS+ and CD235a+ large EV were not statistically elevated in T2D, and large EV did not correlate with the circulating levels of HbA1c (R = -0.384; p>0.055 after adjustment for age and sex). PS+ and CD235a+ large EV were not associated with dyslipidemia, HTN or obesity as cardiovascular risk factors, nor with vascular damage in T2D patients (S3 Table).” 

Please, see S3 Table in Supporting Information, and the matching legend page 21. 

4. Financial disclosure: 

We improved our financial disclosure statement by replacing our sentence « The funders had no bearing on the conduct of the study », with the proposed phrase below on page 20 at paragraph ‘Funding’: 

« The funders had no role in study design, data collection and analysis, decision to publish, or preparation of the manuscript. » 

5. Captions for your Supporting Information files: 

We improved the captions for Supporting Information. This includes titles of the different sections, and expanded captions for the 2 tables. Please, see page 21 at end of manuscript. 

Submission: 

The Figure files were uploaded to the PACE digital diagnostic tool and seemed appropriate. 

Reviewer comments and requirements 

Reviewer #2 quote: [,,,] I have only one concern: in the methods the authors write the following: « We did not attempt to specify the different forms of ferrous, ferryl and deoxy-Hb which are all expected to contribute to Abs575. » 

Pathological consequences of IVH are mainly due to the production and actions of oxidized Hb forms. I am missing the description of Hb oxidation, and the specific pro-oxidant and pro-inflammatory actions of the different products (metHb, ferrylHb, covalently crosslinked Hb forms, free heme). I understand that measuring the concentration of these products are quite challenging and it was not the goal of the current work, but this should be discussed properly to encourage further work in this field.

We addressed this concern by incrementing the discussion as suggested, beyond the notions already introduced. We inserted a new paragraph discussing more specifically the different forms of Hb oxidation products, and listing their specific effects on blood vessels. Please, see l3 last paragraphs of the discussion; page 18 (bottom) and 19 (top), as well as the additional, recent reference by [Bozza et al, Front Immunology, 2020] necessary to support the description of Hb oxidation (Reference #49 Bozza). 

We also added a paragraph pointing at the future possibilities of quantifying the relative Hb degradation products specifically. 

“ These products of Hb oxidation can stimulate specific pro-oxidant and pro-inflammatory actions on the vascular endothelium, with functional consequences. MetHb triggers the endothelial production of interleukin-6 and -8. FerrylHb can trigger endothelial activation especially through Nuclear Factor-kappa B (NF�B) pathway. Globin-derived radicals activate endothelial interleukin-1ß production via the NOD-like receptor family, pyrin domain containing 3 (NLRP3) inflammasome (49). In addition, oxidized forms of Hb shed their prosthetic heme more readily. Some suppose that protein-free heme could trigger specific oxidative reactions in the bloodstream, like hydroxyl radical formation through Fenton reaction or lipid peroxidation (49). However, opportunities for heme to maintain itself in a ‘free’ state are unlikely and would be extremely brief. The odds are that heme will rapidly be exchanged and interact with proteins present in plasma at high concentrations, like lipoproteins, hemopexin or albumin. 

It would be advantageous to know which forms of oxidized Hb and derivatives prevail in T2D plasma in order to imagine therapeutic strategies for their neutralization. Future studies could attempt to distinguish Hb oxidation products in T2D, possibly via absorbance spectroscopy with spectral deconvolution techniques (26,50) or via reverse-phase high performance liquid chromatography (RP-HPLC) (34) and quantify their relative proportions. ” 

In conclusion, we are glad that you found our findings relevant, and hope that the new format and additions make the manuscript suitable for publication in PLOS One. 

With our best regards, 

Olivier Blanc-Brude, Ph.D.

---

## [Editor Report · Decision Letter 1]

14 Sep 2022

Low Grade Intravascular Hemolysis Associates with Peripheral Nerve Injury in Type 2 Diabetes

PONE-D-22-13587R1

Dear Dr. BLANC-BRUDE,

We’re pleased to inform you that your manuscript has been judged scientifically suitable for publication and will be formally accepted for publication once it meets all outstanding technical requirements.

Kind regards,

Alok Raghav, PhD

Academic Editor

PLOS ONE
---

## [Editor Report · Acceptance letter]

26 Sep 2022

PONE-D-22-13587R1 

Low Grade Intravascular Hemolysis Associates with Peripheral Nerve Injury in Type 2 Diabetes 

Dear Dr. Blanc-Brude:

I'm pleased to inform you that your manuscript has been deemed suitable for publication in PLOS ONE. Congratulations! Your manuscript is now with our production department. 

Kind regards, 

on behalf of

Dr. Alok Raghav 

Academic Editor

PLOS ONE